# Analysis of the Internal Mounting Forces and Strength of Newly Designed Fastener to Joints Wood and Wood-Based Panels

**DOI:** 10.3390/ma14237119

**Published:** 2021-11-23

**Authors:** Łukasz Krzyżaniak, Tolga Kuşkun, Ali Kasal, Jerzy Smardzewski

**Affiliations:** 1Department of Furniture Design, Faculty of Forestry and Wood Technology, Poznan University of Life Sciences, Wojska Polskiego 28, 60-637 Poznan, Poland; 2Department of Woodworking Industrial Engineering, Faculty of Technology, Muğla Sıtkı Koçman University, Muğla 48000, Turkey; tolgakuskun@mu.edu.tr (T.K.); alikasal@mu.edu.tr (A.K.)

**Keywords:** fasteners, wood, wood-based boards, corner joints, mounting force, strength, stiffness

## Abstract

This study aimed to numerically and experimentally analyze the effects of internal mounting forces and selected materials on the stiffness and bending moment capacity of L-type corner joints connected with novelty-designed 3D printed fasteners. The experiments were carried out using medium-density fiberboard, high-density fiberboard, beech plywood, particleboard, and beech (*Fagus silvatica* L.) wood. The results showed that the joints made of beech wood were characterized by the largest bending moment capacity (12.34 Nm), while the worst properties were shown by particleboard (2.18 Nm). The highest stiffness was demonstrated by plywood joints (6.56 kNm/rad), and the lowest by particleboard (0.42 kNm/rad). Experimental studies have reasonably verified the results of numerical calculations. The test results confirmed that the geometry of new fasteners promotes the mounting forces under the assembly of the joints. It was shown that the higher the density of the materials, the greater the value of the mounting forces (164 N–189 N).

## 1. Introduction

Any restrictions resulting from anthropometry, disability, age, or place of residence of people significantly impact the availability of services and objects in public spaces, work environments, or apartments. Referring to accessibility, O’Neill pointed out that respect and care are important elements of healthcare. Still, we should also develop the necessary competencies and knowledge to expand accessibility areas [1,2]. Hrovatin et al. [3] studied the functional accessibility of kitchen furniture for users at the age of 65. However, the structure of this furniture and the possibility of assembling it without using tools were not analyzed. Ready-to-assemble (RTA) products, especially furniture, comprise many elements and fasteners. From the literature, it has been known that joints exert a significant influence on furniture durability [4,5,6,7,8,9,10]. Their stiffness and strength depend mainly on the mechanical properties of materials, the number of fasteners, arrangement, dimensions, etc. The most popular separable fasteners used for cabinet furniture are eccentrics and screws. The number and dimensions of the connectors used in these fasteners depend on the strength of the material and the withdrawal strength of screws [11]. The withdrawal strength of screws in plywood and wood-plastic composites was also tested [12,13,14]. In the paper [15], the authors detected localized density effects in wood-based panels on the holding capacities of fasteners commonly used in furniture. They were using static and cyclic tests of the withdrawal and head pull-through of screws and staples and the lateral resistance of screws in oriented strand board (OSB), medium-density fiberboard (MDF), and particleboard (PB). Additionally, Bal et al. [16] determined the direct screw withdrawal and screw head pull-through of selected wood-based materials. Many experiments have also been carried out using wood and wood-based composites such as particleboard, medium-density fiberboard, high-density fiberboard, or oriented strand board to examine the mechanical properties of such type of fasteners [17,18,19]. Langová and Joščák [20] designed and determined the mechanical properties of confirmat screws corner joints. The joints were made of native wood and wood-based composites. Šimek et al. [21] investigated the effect of the end distance of cam lock fasteners on the bending moment resistance of knock-down corner joints. In work [22], the authors presented the effect of moisture content on the mechanical properties of corner furniture joints when different joining methods and materials were used. It was detected that when the same materials were bonded, maximum load carrying capacity was achieved with PUR adhesive for the material combination of plastic-plastic and moisture content of 90%. New types of adhesives and fasteners were also studied [23,24,25,26].

A characteristic feature of the tested joints was the need to use the external energy required to generate mounting forces. These forces were most often applied to screws or eccentrics in the form of screwing moments, usually with screwdrivers [27,28,29]. In the paper [30], the furniture screws and fasteners and the moments of their drive-in values on the strength and durability of drawer runners were investigated. The authors calculated the optimal driven-in screws moment during the mounting of fasteners and hinges to the furniture body.

The popularity of furniture in flat packages RTA, however, forces a new need to expand their availability among the elderly, disabled, or those who do not have the technical ability to assemble with tools. For this reason, new fasteners are created with an original design and method of operation [25,31,32]. New joints are externally invisible, quickly mounted, and dismounted without tools. Common tests of this type are computer simulations using the finite element method (FEM) [25,31,33,34,35]. From a cognitive and practical point of view, it would also be interesting to determine the impact of the mechanical properties of wood-based materials on the mounting forces and stiffness of newly designed joints for RTA furniture. Research to date in this area has mainly related to the joints of chest furniture assembled using tools. Therefore, the design of the new fasteners, unlike the ones described above, requires that fasteners must be mounted in the furniture elements in a factory. Therefore, the joints should be easy to assemble without any mistakes by employees. They should be inserted into previously made sockets/holes by easy pressing. The shape of fasteners should be symmetrical and interchangeable, without the need for positioning relative to other elements or holes. Besides, the strength of furniture joints made with these fasteners should depend on the type of material used. However, this impact should not disqualify any of the materials.

In the context of the presented literature, limited studies have raised the problem of the relationship between the strength of furniture joints and the internal forces resulting from the assembly of elements. Furthermore, there have also been no studies explaining how the geometry of the fasteners and the type of materials used for their production will affect mounting forces. Consequently, the relationship between the mounting forces in the newly designed fasteners and the stiffness of these joints has not been established. Accordingly, this study decided to design a new original fastener for wood-based composites and cabinet furniture joints that is invisible from the outside and easy to assemble by the consumer without using tools. This study aimed to numerically and experimentally analyze the effects of internal mounting forces and selected materials on the stiffness and strength of L-type corner joints.

The disposition of this study could be expressed in four sections. In Section 1, the shape of the newly designed fastener and the mathematical model and distribution of the internal forces were presented. In Section 2, the preparation of the L-type corner joints connected with this fastener was described. Section 3 and Section 4 include the presentation of testing and numerical analyses of the joints, respectively.

## 2. Materials and Methods

Figure 1 shows a flowchart illustrating the universality of the methodology used. In the first step, the engineer designs and analyzes the construction of fasteners, their geometry, and the possibility of modifying the shape. Next, the method requires collecting the basic elasticity constants of the materials from which the joints are made. The next step involves analytical modeling of the internal mounting forces. This leads to an explanation of where the desired interactions can be expected to occur. Then, the method provides for the parallel conduct of experimental and numerical tests to determine the strength and stiffness characteristics of the joints. Finally, the engineer should compare the relationship between the force and displacement numerically and experimentally measured. The high correlation of this relationship proves the correct calibration of the numerical model. Next, the engineer should analyze the mounting forces in the joint and the interaction between the surfaces of the fasteners. The last stage of implementing the described method is assessing the influence of mounting forces and the type of materials used on the stiffness and strength of furniture joints.

### 2.1. Fastener Model and Distribution of the Internal Forces 

A characteristic feature of the designed fastener is its external invisibility after assembling the furniture. The fastener is composed of two elements, namely, mortise and tenon (Figure 1). The shape of both elements ensures self-assembly without any tools and no mistakes by the user. Assembling the fastener elements in corner joints involves shifting the two panels by 40 mm relative to each other, as illustrated in Figure 2.

During the assembly of the corner joints, mounting force F (N) is created, which causes the mutual pressure of the panels. The mounting force value can be determined by analyzing the distribution of forces in the joint (Figure 3). On each of the tenon and mortise contact surfaces, there are 1/2F (N) forces resulting from the friction force T (N) on the contact surface and the normal force N (N) perpendicular to the contact surface. The force $F_{1}$ (N) is a component of the force N (N) and acts perpendicular to the material’s surface in which the mortise was prepared. Considering this, the value of force $F_{1}$ (N) can be calculated from the system of Equation (1): (1)$$\{\begin{array}{l} F_{1}=N\cos\left(\propto \right)\\ N=\frac{1}{2}F\sin\left(\propto \right) \end{array}$$
hence:(2)$$F_{1}=\frac{1}{2}F\sin\left(\propto \right)\cos\left(\propto \right)$$
where: α (rad) is the angle of tenon and mortise wall inclination. Because the force $F_{1}$ (N) puts pressure on the material of the mortise and the material of the panel in which the mortise is fixed, it will cause linear strains in the direction of its action (Figure 3). The total strain ε will be equal to the sum of strains in the panel $\varepsilon_{w}$ and in the mortise $\varepsilon_{p}$ (Equation (3)):(3)$$\varepsilon =\varepsilon_{w}+\varepsilon_{p}$$

Using Hooke’s law, Equation (3) can be written in the form:(4)$$\frac{F_{1}}{AE}=\frac{F_{1}}{AE_{w}}+\frac{F_{1}}{AE_{p}}$$
where: A (mm^2^) is the vertically projected surface area of contacting elements, $E_{w}$ (MPa), the modulus of elasticity for the panel material, and $E_{p}$ (MPa), the modulus of elasticity material of the mortise. From Equation (4) the modulus of linear elasticity E (MPa) can be determined for a system of serially related materials:(5)$$E=\frac{E_{w}E_{p}}{E_{w}+E_{p}}$$

Finally, using the Equations (2), (3), and (5), the mounting force F (N), in the range of the linear elasticity of materials, can be present in the form:(6)$$F=\frac{2\varepsilon E_{w}E_{p}}{\left(E_{w}+E_{p}\right)\sin\left(\propto \right)\cos\left(\propto \right)}$$

Taking into account the difficulty of experimental determination of the strains $\varepsilon_{w},\;\varepsilon_{w}$, and ε and the fact that these strains may exceed the range of linear elasticity, it was decided to determine the mounting force F (N) based on numerical calculations.

### 2.2. Preparation of the L-Type Corner Joints 

The fastener elements (tenon and mortise) were manufactured of polyamide PA12 in 3D printing technology (selective laser sintering) on the EOS P396 printer (EOS GmbH, Munich, Germany). During the 3D printing, the thickness of the layers was equal 0.2 mm. Each layer was applied with an infill density of 100% towards the long side of the fasteners. Then, L-type corner joint specimens were prepared of five selected materials: medium-density fiberboard (MDF), high-density fiberboard (HDF), beech (*Fagus silvatica* L.) plywood (PL), particleboard (PB) and beech (*Fagus silvatica* L.) wood (Be) (Figure 4) by utilizing the manufactured fasteners. The materials were 18 mm in thickness. The choice of materials was imposed by the fact of their widespread use in the furniture industry around the world. The L-type specimen consists of two panels of the same type of material, joined together by two fasteners. The first panel measured 400 × 100 × 18 mm, whereas the other panel measured 400 × 82 × 18 mm. It was recognized that the designed new fastener could soon be used instead of traditional furniture fasteners to assemble case furniture from the same materials.

The dimensions of the corner joint specimens are shown in Figure 5. The length of the joint arms was 400 mm. In the arms, on the CNC Format 200 milling machine (Folder Group, Żory, Poland), sockets and holes were made using an HM straight shank cutter of 8 and 10 mm in diameter (CMT, Poznań, Poland). Taking into account the diameters of the tools and the dimensions of the fasteners (Figure 1), an interference fit of +0.2 mm was obtained. Fastener elements are inserted into the sockets and holes without glue. Totally, 100 representative L-type corner joint specimens were prepared, including five materials and twenty replications for each material. All of the materials were seasoned in laboratory conditions at a temperature of 20 °C ± 2 °C and a relative air humidity of 60% ± 5% for one month until their mass was constant (moisture content 8%). The densities and some mechanical properties of the materials used are given in Table 1.

### 2.3. Testing of the L-Type Corner Joints

In everyday use, the corner joints of cabinet furniture are exposed to two main forces: tension and compression. Most of these forces are applied through cantilevers (long sides) and can generate sizable bending moments. Figure 5 shows loading diagrams in the testing corner joint.Accordingly, the prepared L-type corner joint specimens were subjected to static tension and compression loads. The experiments were carried out in a Zwick 1445 testing machine (Zwick Roell AG, Ulm, Germany) with a loading rate of 10 mm/min. In the tension tests, the bottoms of each of the two arms of the joints were placed on rollers so that the two joint arms were free to move outwardly as the corner joint was loaded. During the experimental studies, the value of force P (N) was measured accurately to 0.01 N, and displacement in the direction of acting force DP (mm) was determined accurately to be 0.01 mm. A total of 100 L-type specimens were tested; 50 were tested in tension and the remaining 50 in compression. For selected load diagrams, the bending moment capacities (*M_T_, M_C_*) of the joints were calculated from the formulas:(7)$$M_{T}=0.5P_{max}e{}^{\prime }\;(\mathrm{Nm})\;\mathrm{tension},$$
(8)$$M_{C}=P_{max}a{}^{\prime }\;(\mathrm{Nm})\;\mathrm{compression},$$
where: $P_{max}$ (N) is the maximum force destroying the joints, $e{}^{\prime }$ (mm), and $a{}^{\prime }$ (mm) is the length of the arm of the force $P_{max}$ (N). The moment arms ($e^{\prime },\;a{}^{\prime }$) were calculated as 0.058 m from the formulas (11) and (18), respectively, for both loadings. 

The stiffness of the corner joints was calculated as the quotient of the bending moment $M_{T}$ (Nm) and $M_{C}$ (Nm) and the angle φ (rad) between the joint arms. This angle was determined based on the measurement of displacement DP (mm) caused by external load P (N). The coefficient of stiffness $K_{T}$ (kNm/rad) for the joints under tension is given by:(9)$$K_{T}=\frac{Pe^{\prime }}{2\varphi }$$
where:(10)$$\varphi =\left(\varphi_{2}-\varphi_{1}\right)$$
(11)$$e^{\prime }=\frac{\sqrt{2}}{2}\left(a-b\right)$$
(12)$$0.5\varphi_{1}=atg\left(\frac{e{}^{\prime }}{f}\right)$$
(13)$$0.5\varphi_{2}=atg\left(\frac{e{}^{\prime \prime }}{f-DP}\right)$$
(14)$$f=e^{\prime }+\frac{\sqrt{2}}{2}b$$
(15)$$e^{\prime \prime }=\sqrt{e^{\prime 2}+f^{2}-{\left(f-DP\right)}^{2}}$$

P (N) is the external load, DP (mm) is the displacement in the direction of the force P (N), a, b (mm) are the dimensions of the joint arms, $e^{\prime }$ (mm) is the bending moment arm, f (mm) is the specimen height, and $a^{\prime }$ (mm) is the bending moment arm, as in Figure 4. The stiffness coefficient $K_{C}$ (kNm/rad) for the compressed joints is given by the equation:(16)$$K_{C}=\frac{Pa^{\prime }}{\varphi }$$
where:(17)$$\varphi =\left(\varphi_{1}-\varphi_{2}\right)$$
(18)$$a^{\prime }=\frac{\sqrt{2}}{2}a-a^{\prime \prime }$$
(19)$$a^{\prime \prime }=b\sqrt{2}$$
(20)$$\varphi_{1}=2atg\left(\frac{\sqrt{2}a}{2a{}^{\prime }}\right)$$
(21)$$\varphi_{2}=2asin\left(\frac{\frac{\sqrt{2}}{2}a-DP}{\sqrt{b^{2}+{\left(a-b\right)}^{2}}}\right)$$

### 2.4. Numerical Model of the Corner Joints

In this study, numerical calculations were made using the Abaqus/Explicit v.6.13-1 software (Dassault Systemes Simulia Corp., Waltham, MA, USA). Computations were performed at the Poznań Supercomputing and Networking Center (PSNC) using the Eagle computing cluster. 

In preparing to develop the most accurate and uncomplicated numerical model of L-type corner joints geometry, the kinematics of joints assembly were analyzed in detail. It was assumed that 3D joint models made in Autodesk Inventor 2020 are fully compatible with the prototypes made on their basis. This assumption was positively verified by comparing the dimensions of the manufactured tenon and mortises with the dimensions in the technical drawings; on this basis, the 3D model in 1:1 scale as *.STP files was prepared. These files were imported into the program Abaqus/Explicit v. 6.13-1 (Dassault Systemes Simulia Corp., Waltham, MA, USA) [38,39]. Then the kinematics of joint assembly were thoroughly analyzed, marking which surfaces of tenon and mortise and which surfaces of joint arms have sequential contact with each other. It has been observed that, firstly, contact should be assigned between the wide and narrow surface of the joint arms. Then, contact between the surfaces of mortise and tenon should be assigned. Since the tenon has a width of 0.2 mm larger than the width of the mortise, it was considered appropriate to take into account in the numerical model both the friction between the fasteners’ surfaces and the contact stress resulting from pressing the tenon into the mortise during uniform motion. Thanks to the solutions applied, it was decided to induce mounting forces in the joints and the resulting contact stresses on the surfaces of the joint arms. The models of joints with mounting forces prepared in this way were loaded to compression and tension in accordance with the methodology given in point 2.3. 

Given the above, firstly, numerical models were prepared to calculate the value of the mounting forces and contact pressures in corner joints, depending on the type of material from which the joint arms were made. The elastic properties of polyamide PA12, wood, and wood-based materials were collected in Table 1. For all contact surfaces between the moving parts of the joints, the values of the friction coefficient 0.2 were assigned. The interfacial bonding between the connector and wooden materials was assumed to be perfectly bonded. In general, an 8-node linear brick, reduced integration, hourglass control element C3D8R was used (about 50,200 elements and 60,300 nodes per model). PB, MDF, HDF, and PA were modeled as elastic-perfectly plastic materials, while PL and Be were modeled as elastic-orthotropic material. In addition, geometric nonlinearity was considered to represent the large deformation of the structure. After mapping the assembly process of joints, three steps were defined for boundary conditions and loads. In Step 1, the 15 mm shift towards the axis X (1) of the vertical member over the surface of the horizontal member was used (Figure 6a). Displacement was defined at the RP1 reference point of the tenon (Figure 6d). During this time, the possibility of displacement in other directions and rotations relative to the X, Y, Z axes was fixed. This displacement allowed for the first contact between the tenon and mortise at the reference point RP2 (Figure 6d). In Step 2, at point RP1, further displacement in the direction of the axis X (1) by 25 mm was forced, and, additionally, the displacement of the vertical element in the direction of the axis Z (3) was made possible. As a result, the process of assembling the joint was restored, and the position was obtained as in Figure 6b. During these two steps, the values of the mounting force F (N) at the RP1 reference point and the change of the contact pressures in point RP2 of the mortise and tenon were calculated.

Subsequently, in the simulation introduced Step 3, the boundary conditions were changed, as in Figure 4. A local coordinate system was created, and DP (mm) displacements were applied to cause the joint tension and compression (Figure 4). At the same time, the mounting forces F (N) active in Step 2 were active. Based on the results of the calculations in Step 3, the reaction forces P (N) causing specific displacements DP (mm) were determined. Ten appropriate models for each type of material and method of loading (tension and compression) were prepared.

The quality of the developed FEM models was evaluated by a comparison of the load-displacement curves obtained from the results of experiments and numerical calculations. If the adequate curves coincide, the numerical models have been correctly calibrated, and the numerical analysis results are adequate. 

## 3. Results

### 3.1. Fastener Model and Distribution of the Internal Forces 

The quality of calibration of the developed FEM models of L-type corner joints was evaluated by a comparison of the load-displacement curves obtained from the results of experiments and numerical calculations. Figure 7 illustrates the experimental (average values) and numerically calculated dependence of the load on the displacement of corner joints. This figure shows that the results of the numerical calculations were highly satisfactory in both cases, confirmed by the experimental results. It means that the adopted numerical models leading to the determination of mounting forces are correct. 

This conclusion was also confirmed by an appropriate comparison of the maximum load values. Table 2 summarizes the maximum destructive loads determined experimentally and numerically. This table shows that, under both tension and compression, the maximum load differences of the joints are not greater than 7% and range from −6.96% to 4.12%. This means that the influence of mounting forces and the type of material on the stiffness and strength of the corner joints was correctly mapped. So, further analysis based on numerical calculations is correct and leads to the right conclusions.

### 3.2. Effect of the Material Type on Mounting Forces of Corner Joints

Figure 8 presents the results of numerical calculations over the course of the assembly of the joints. As it results from the arrangement of individual curves, the mounting force irregularly changes the value along the length of contact between the tenon and mortise (Figure 8a). The value of maximum mounting force for individual types of PL, HDF, MDF, Be, and PB materials is 189 (N), 189 (N), 179 (N), 172 N, and 164 (N), respectively. This comparison shows that, in the joints constructed of PL and HDF, the mounting force is 5.9%, 10%, and 15.5% higher than the forces occurring in the joints constructed of MDF, Be, and PB. The main reason for this is the highest density of PL and HDF, respectively 798 and 891 kg/m^3^. In other cases, the density for MDF, Be, and PB was 745, 734, and 642 kg/m^3^. Based on this discussion, it can be concluded that the connection material has a significant impact on the mounting forces.

Figure 8b shows that the joints constructed of HDF are distinguished by the highest value of contact pressure occurring in the interaction between the surface of the tenon and mortise. The reason for this is the previously mentioned highest density of HDF boards used. This is directly related to the physic-mechanical properties of the materials used. The method of mounting the connections causes a significant expansion of the mortise through the tenon. Therefore, materials with a higher modulus of linear elasticity ($E_{w}$) experience smaller deformations caused by compression, which changes the contact stress distribution on the contact surface of the mortise and tenon. This phenomenon is illustrated below.

Figure 9a shows that the mounting forces caused by assembly also contribute to the formation of contact pressure on the surface of contact elements. The largest of these pressures occurred around the fasteners and, in some places, on the surface of the panels. Such an uneven distribution of contact pressures on the surface of the boards resulted from the irregular work of the abrasive connectors during assembly. Furthermore, the mutual pressure of the tenon and mortise surfaces during the joint assembly causes the mortise arms to open (Figure 9b). This figure shows that the maximum reduced stresses are no more than 20 MPa and are 6 MPa smaller compared to the strength of PA12 (Table 1). The above results indicated that the new fasteners correctly produce the expected effects in mounting forces of significant value. The fasteners are also characterized by the sufficient strength of the structure obtained in 3D printing technology.

### 3.3. Effect of the Material Type on Stiffness and Bending Moment Capacity of Corner Joints

The differences in the stiffness of joints are illustrated in Figure 10. It presents the variability of the mean stiffness coefficient as a function of the angle of the joints under tension and compression loads. This figure shows that the stiffness of the joints under tension (Figure 10a) is at least three times higher than the stiffness of the joints under compression (Figure 9b). In addition, in the first phase of loading, i.e., until angular deformation φ = 0.0005 rad, the joint stiffness increases significantly. Then, asymptotically, it gets lower as the angular deformations increase. At the same time, it can be seen that the best mechanical properties are shown by the joints constructed of PL and Be, followed by HDF, MDF, and PB. 

One-way analyses of variances (ANOVA) general linear model procedure were performed for both the bending moment capacity and the stiffness data of L-type corner joints in order to analyze the effect of material type on mean bending moment capacity and stiffness under tension and compression loading. The analyses of variances are given in Table 3 for the bending moment capacity and the stiffness of joints under both loadings. 

The ANOVA results indicated that the effect of material type on the bending moment capacity and the stiffness of L-type corner joints were statistically significant at 5% significance level under both loading types. Therefore, the least significant difference (LSD) multiple comparisons procedure at 5% significance level was performed to determine the mean differences of bending moment capacity and stiffness values of the L-type joints tested. The mean bending moment capacity and mean stiffness of the joints are compared in Figure 11. Standard deviations are shown in this figure as black lines, and the values followed by the same capital letter in parentheses are not significantly different. 

Figure 11 shows that the joints’ bending moment capacities and stiffness are significantly affected by the material type. According to Figure 11a, the highest bending moment capacity is characterized by the joints constructed of Be (12.34 Nm) under tension. The bending moment capacity of joints constructed of PL, HDF, MDF, and PB is lower by 9.3%, 46.2%, 59.2%, and 67.1%, respectively. In the case of the joints under compression, the trend is analogous. The joints constructed of Be (8.57 Nm) are characterized by the highest bending moment capacity. However, the bending moment capacities between the joints constructed of Be and PL are not statistically different. The bending moment capacity values for the joints constructed of Be averaged only 5.2% greater than those for the joints constructed of PL. The bending moment capacity of joints constructed of HDF, MDF, and PB is lower by 47.4%, 67.5%, and 74.6%, respectively. The largest mounting forces presented for PL and HDF and their differences in relation to other materials are not significant enough to have an impact on the bending moment capacity of L-type corner joints. Therefore, the bending moment capacity of the corner joints is determined by the elastic constants of materials, particularly the linear elasticity modules, listed in Table 1.

The grouping data for joints subjected to a compression loading test yielded a mean bending moment capacity of 5.34 Nm, while the grouping data of joints tested in tension loading resulted in a mean bending moment capacity of 7.85 Nm. Therefore, in general, it can be deduced that the joints loaded in tension have greater bending moment capacities than those loaded in compression. The bending moment capacities for the joints loaded in tension averaged approximately 47% greater than those for the joints loaded in compression in this study.

The bending moment capacity results obtained from this study have been compared with the results in some similar studies in the literature [4,21,24,27,28,32]. In these studies, the bending moment capacities of different kinds of corner joints constructed of PB or MDF were investigated by many researchers. 

In the study carried by Jivkov et al. [4], bending moment capacity was obtained as 7.73 Nm for PB-minifix joints, while it was obtained as 9.91 and 14.24 Nm for the same combination in the studies of Šimek et al. [21] and Yerlikaya [24], respectively. In these studies, the bending moment capacity value of glued-dowel and unglued dowel joints was 20.94 and 14.1 Nm. Accordingly, it is understood that the use of glue in the joints has a significant effect on the strength. According to the results of the studies investigating the screwed corner joints [27,28], it can be clearly seen that screw connections provide much higher bending moment capacity values than the other fasteners. The bending moment capacities of glued-screwed and un-glued screwed joints were given as 87.98 and 70.92 Nm in the mentioned studies. Here, it should be noted that, from the point of view of the engineering design approach, the strength of the joints is only related to the loads they must carry in service. The bending moment capacity values obtained in the study [32] in which two different newly externally invisible fasteners were designed and their performance was measured were 19.75, 19.53, 7.13, 7.40, 8.66, and 9.62 Nm for the joints PB-fastener1 (S-PB), MDF-fastener1 (S-MDF), PB-fastener2 (M-PB), MDF-fastener2 (M-MDF), PB-minifix (E-PB), and MDF-minifix (E-MDF), respectively.

When the bending moment capacity values obtained from these aforementioned studies are compared with the bending moment capacity values of this study, it can be seen that there is consistency between the results. However, it is expected that there will be some differences between the bending moment capacity values due to the differences in L-type specimen sizes and corresponding moment arms. The fact that the results obtained from this study are compatible with different joining techniques tested in other studies is an indication that the newly designed fastener can be an alternative for the corner joints of case furniture.

It is clear that from all of those studies, beech and plywood generally yielded higher bending moment capacity results than the other wood-based panels. Similarly, MDF showed higher results than PB, and screwed joints gave higher bending moment capacities than the other joints tested. When the studies are observed in the literature, it can be seen that bending moment capacities under tension and compression are in a linear relationship with the density of the materials used.

Figure 11b illustrates the differences between maximum stiffness coefficients. It follows that the highest stiffness is characteristic of the joints constructed of PL (6.56 kNm/rad) under tension. The stiffness of the joints constructed of Be, HDF, MDF, and PB is lower by 10.8%, 48.5%, 55.1%, and 67.2%, respectively. In the case of the joints under compression, the trend is also analogous. The joints constructed of PL (1.56 kNm/rad) have the highest stiffness. The stiffness of the joints constructed of Be, HDF, MDF, and PB is lower by 9.4%, 50.2%, 59.3%, and 73.0%, respectively. Furthermore, in this case, the high stiffness of the joints does not result from the value of mounting forces but depends on the solid elastic materials (Table 1). 

The primary failure mode of the joints was sliding the tenon out of the mortise (Figure 12a,b). This phenomenon was also obtained during the numerical calculations. As this effect was characteristic of all joints and materials, in Figure 12c,d only one failure case was shown and discussed. The damage is accompanied by significant stresses in the fasteners and not in the joined materials. The largest stresses (24.8 MPa) are concentrated in the base of the cylinder-cone shape of the fastener and are close to the strength of PA12. In the joints’ arms, the stresses do not exceed 17.5 MPa. For wood, plywood, MDF, and HDF, these stresses do not exceed the bending strength of these materials (Table 1). Only in the case of particleboard can the damage of the hole be observed as the strength of the material used, equal to 12 MPa.

## 4. Conclusions

The test results confirmed that the geometry of fastener elements promotes the formation of significant mounting forces under the assembly of the joints. The value of these forces slightly depends on the type of materials used to build the joints. However, it can be seen that the higher the density of materials, the greater the value of these forces. In the joints constructed of beech plywood and HDF, the mounting force is 5.9%, 10%, and 15.5% higher compared to the forces occurring in the joints constructed of MDF, beech wood, and particleboard, respectively. They prevent the connectors from sliding out of their mortises. Similar values of mounting forces justify the conclusion that the stiffness and strength of joints depend on the type of materials used, especially the values of linear elasticity modules. The bending moment capacity of the corner joints is determined by the elastic constants of materials, particularly the modules of linear elasticity. The highest bending moment capacity of 12.34 Nm is characterized by the joints constructed of beech wood under tension and 8.57 Nm under compression. In joints constructed of beech plywood, HDF, MDF, and particleboard, the bending moment capacities are lower, from 9.3% to 67.1%. The joints are characterized by high initial stiffness recorded for angular deformation *φ* = 0.0005 radians. In the case of the joints constructed of plywood, the highest coefficient of stiffness is equal to 6.56 kNm/rad under tension and 1.56 kNm/rad under compression. All of the results from experimental studies showed consistency in the results of numerical calculations. This indicates that the described phenomena of excitation mounting forces and the nature of the interaction between the fastener elements in the joints are correct.

This research has shown the usefulness of 3D printing for the production of experimental fasteners for furniture joints. In industrial practice, however, it is proposed to use cheaper injection technology. This fastener is able to mount and dismount the furniture elements on-site without tools quickly. They are easily put into previously made sockets/holes by automatic insertion. Furthermore, the shape of fasteners is symmetrical and did not need any positioning relative to other elements or holes.

The presented method applies only to joints in which mounting forces are generated. Therefore, its use in the case of adhesive joints will not be effective.

A new direction of research using the developed method will be the implementation of metamaterials with a negative Poisson’s ratio for modeling increasing mounting forces in furniture joints.

## Figures and Tables

**Figure 1 materials-14-07119-f001:**
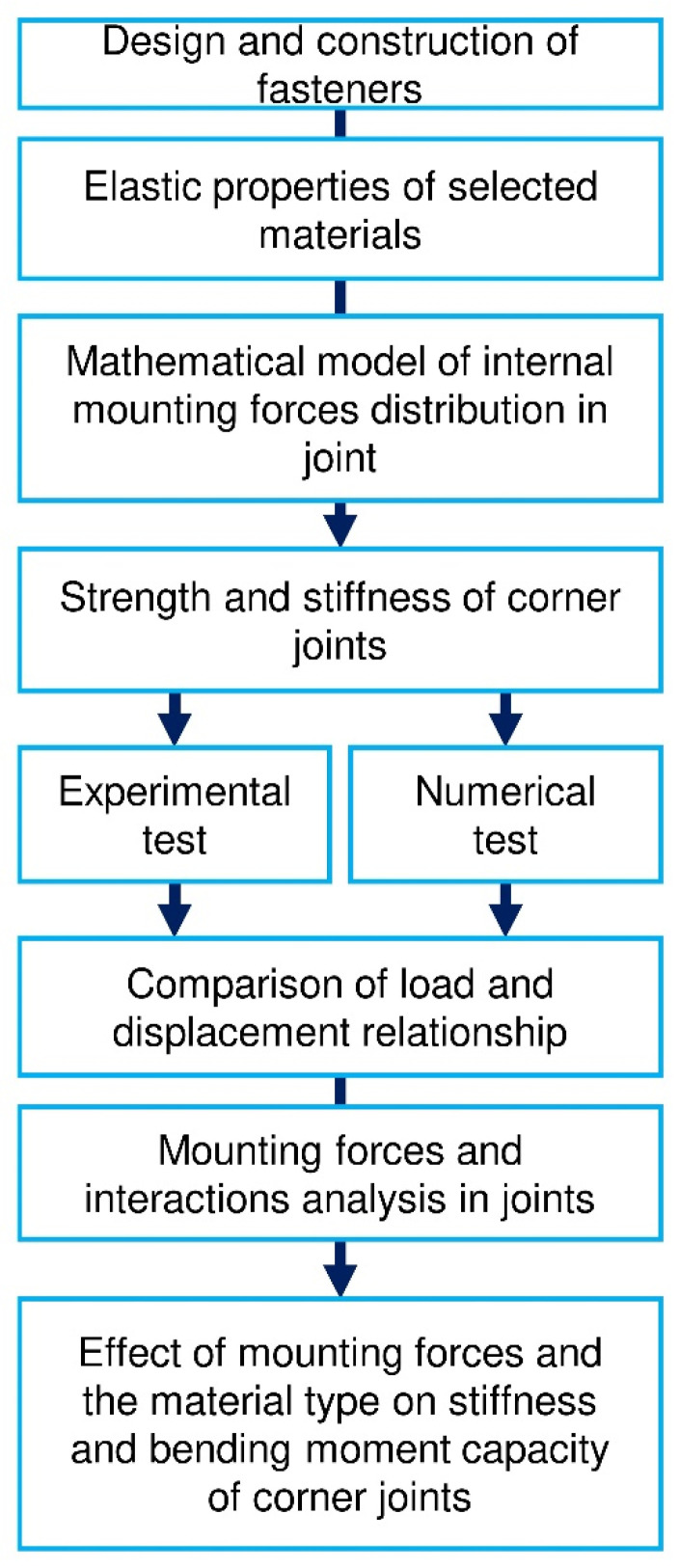
The flowchart of the used methodology.

**Figure 2 materials-14-07119-f002:**
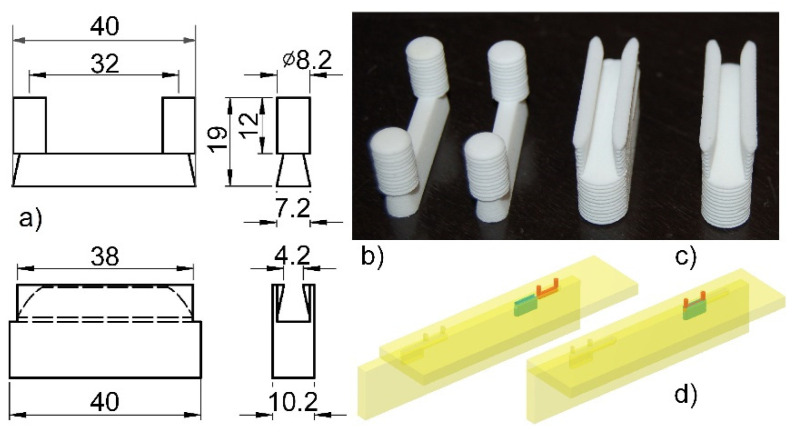
Configuration of the designed mortise-tenon fastener system and assembling: (**a**) dimensions (in mm), (**b**) mortise, (**c**) tenon, (**d**) joint before and after mounting.

**Figure 3 materials-14-07119-f003:**
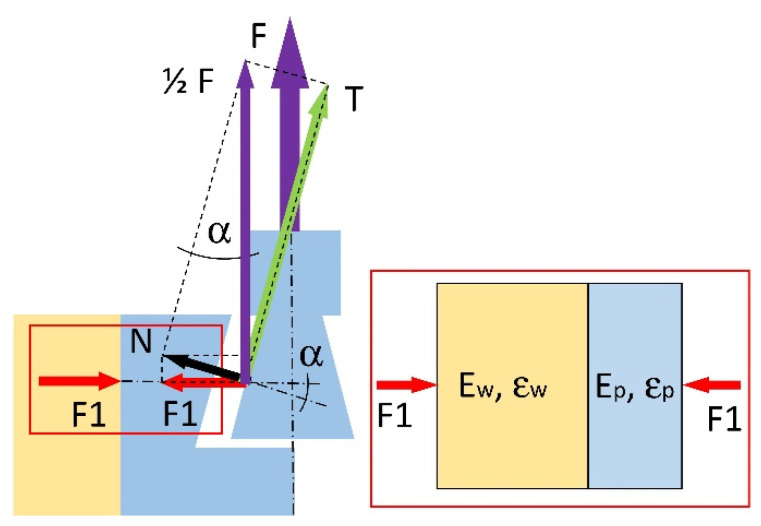
Distribution of the internal forces in the designed fastener: F (N) mounting force, T (N) friction force, N (N) normal force, $F_{1}$ (N) component of the force N (N), α (rad) angle of tenon and mortise wall inclination, ε total strain, $\varepsilon_{w}$, $\varepsilon_{p}$ strains in the panel and mortise, $E_{w}$ (MPa) modulus of elasticity for the panel material, $E_{p}$ (MPa), the modulus of elasticity material of the mortise.

**Figure 4 materials-14-07119-f004:**
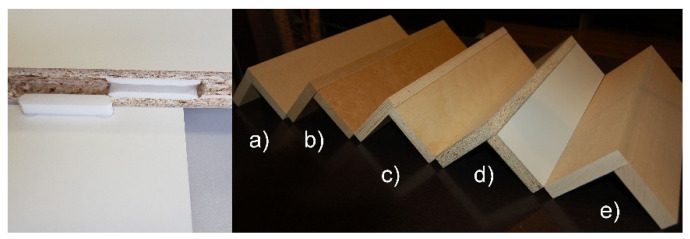
L-type corner joint specimens constructed of: (**a**) MDF, (**b**) HDF, (**c**) PL, (**d**) PB, (**e**) Be.

**Figure 5 materials-14-07119-f005:**
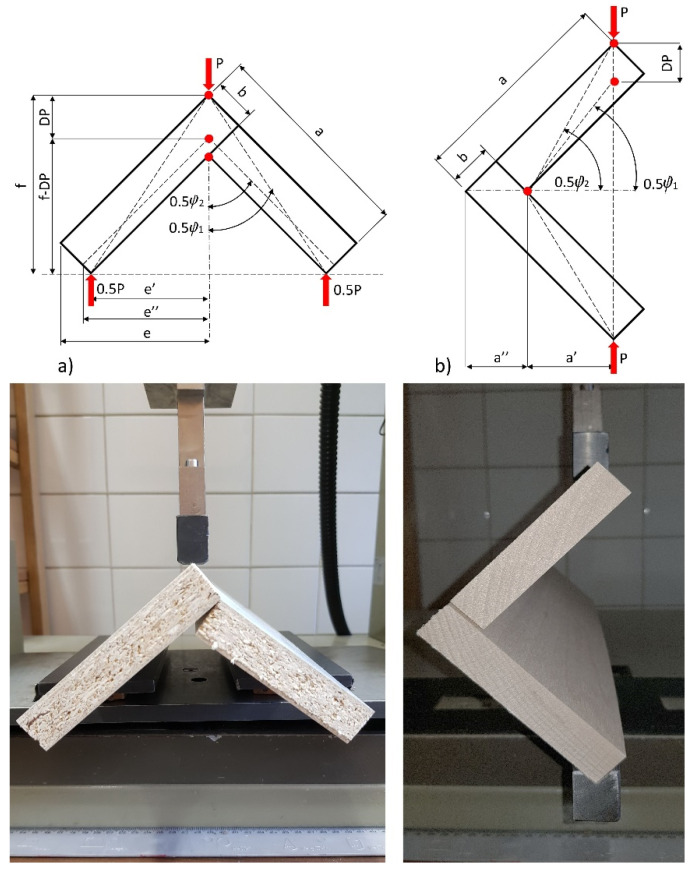
Loading of the corner joints: (**a**) under tension, (**b**) under compression (a = 100 mm, b = 18 mm).

**Figure 6 materials-14-07119-f006:**
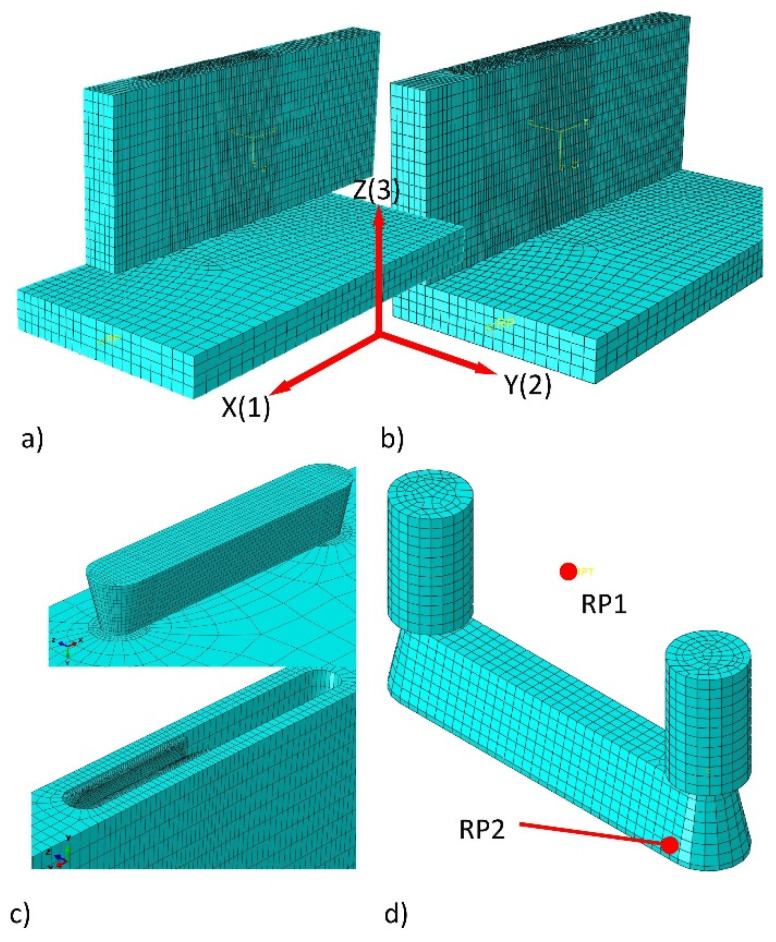
A numerical model for the simulation of mounting forces and the stiffness of joints: (**a**) mesh model of joint before assembly, (**b**) mesh model of joint after assembly, (**c**) tenon and mortise, (**d**) tenon with reference points.

**Figure 7 materials-14-07119-f007:**
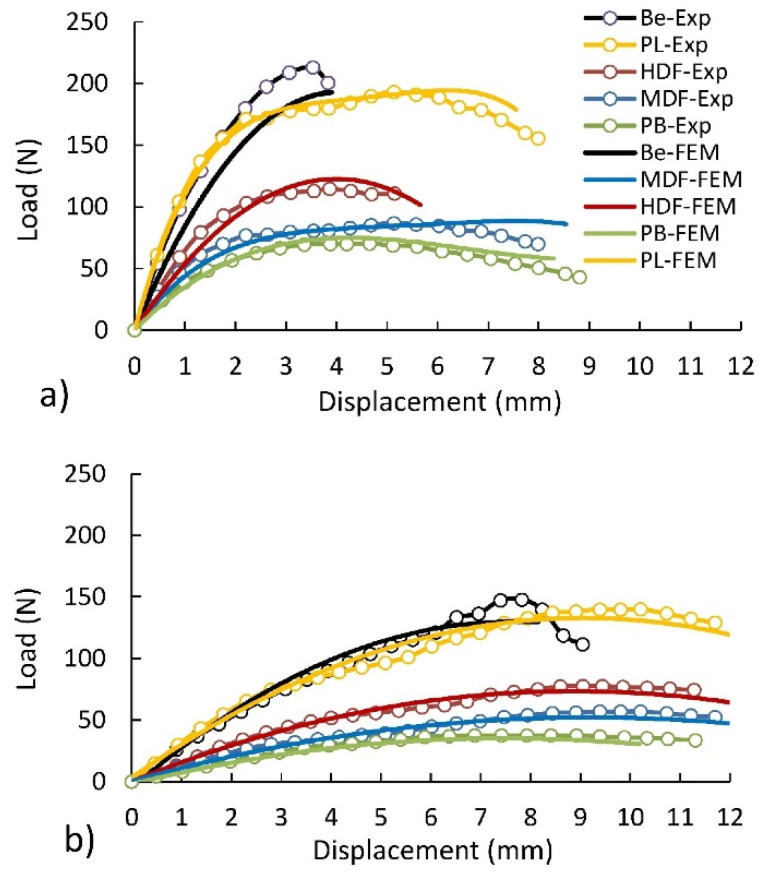
Load dependence on displacement for joints: (**a**) under tension, (**b**) under compression.

**Figure 8 materials-14-07119-f008:**
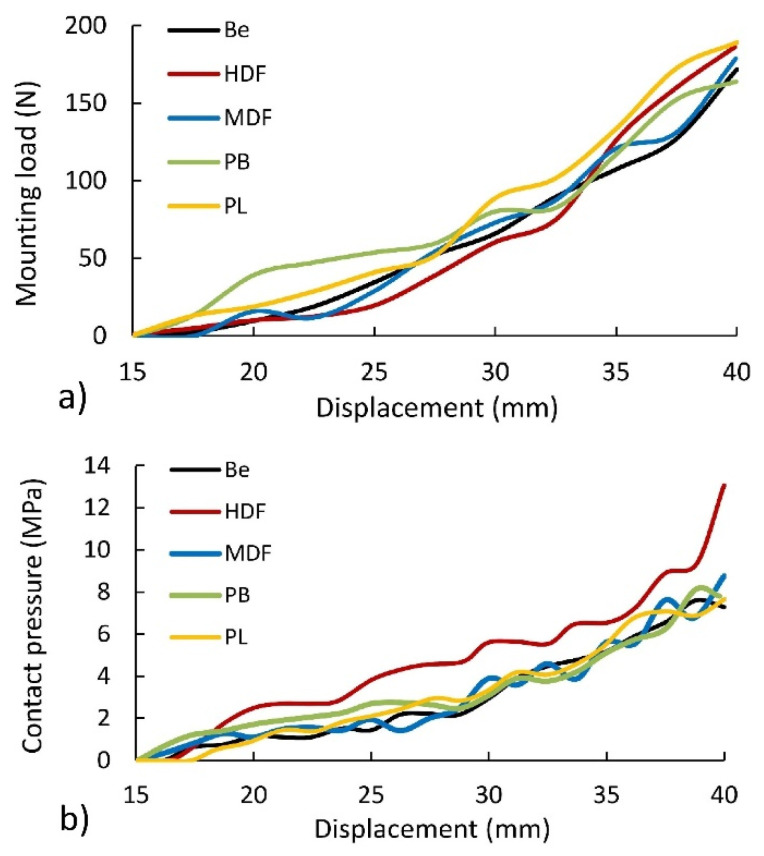
Dependence of mounting force (**a**) and contact pressure (**b**) for material type.

**Figure 9 materials-14-07119-f009:**
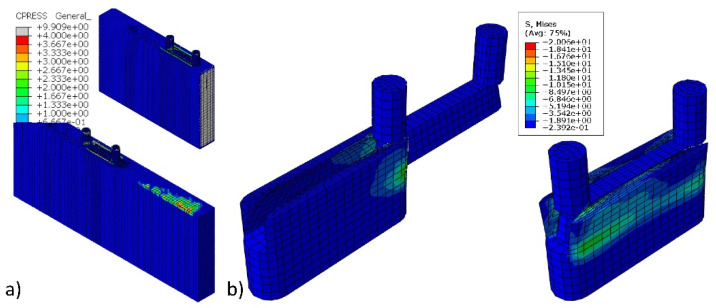
Contact pressures (**a**) and reduced stress distribution (**b**) during the assembly of joints.

**Figure 10 materials-14-07119-f010:**
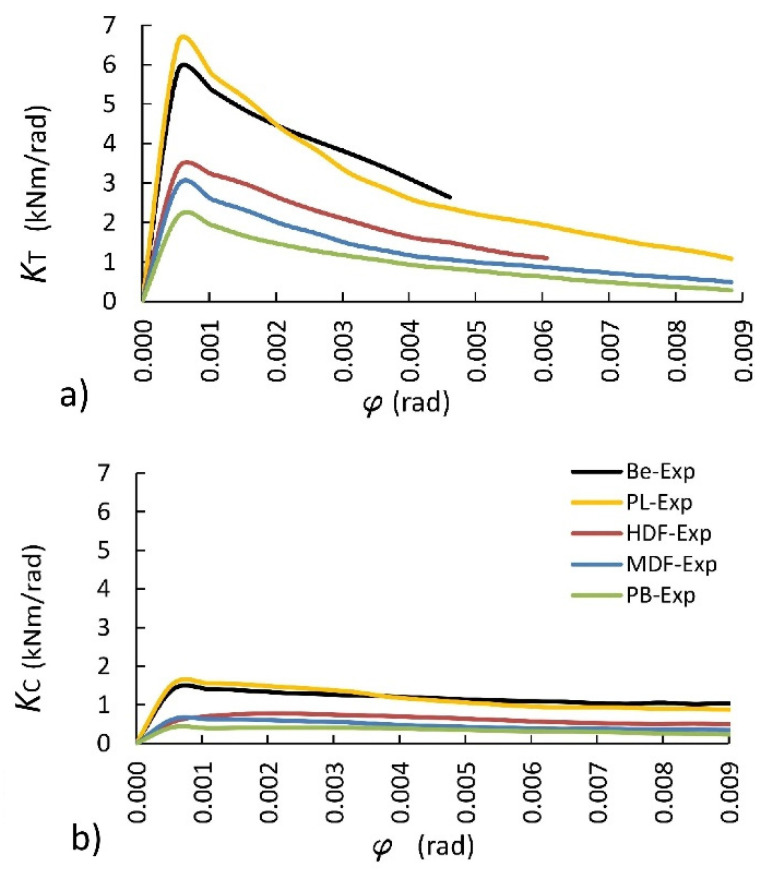
Relationship between the stiffness coefficient and the deformation angle: (**a**) under tension, (**b**) under compression.

**Figure 11 materials-14-07119-f011:**
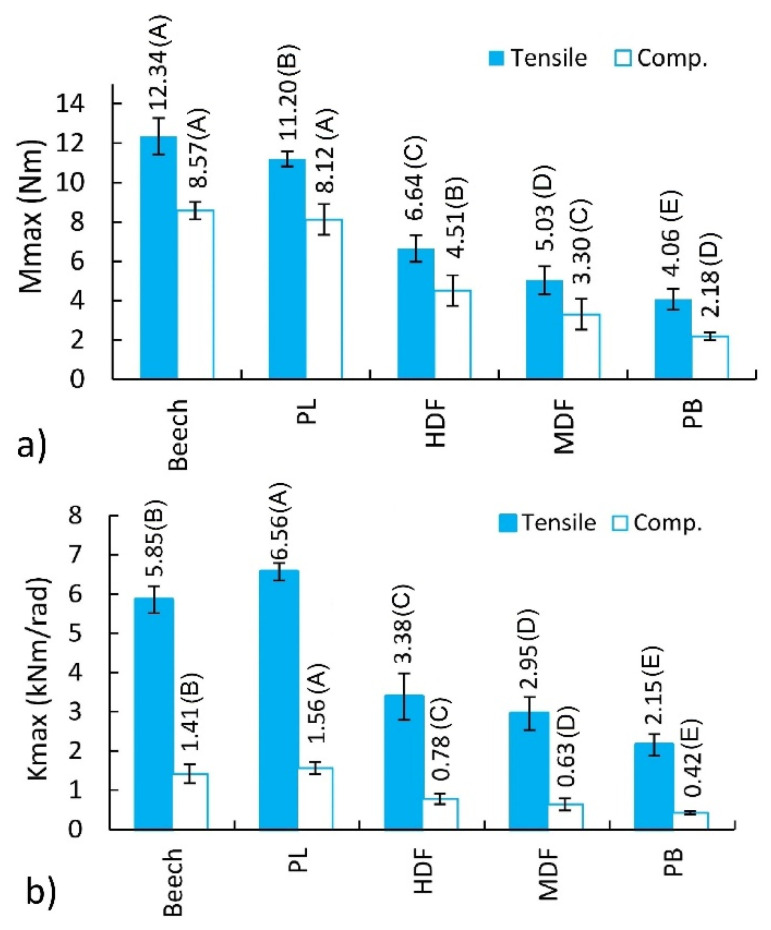
Bending moment capacities (**a**) and stiffness (**b**) of joints under tension and compression.

**Figure 12 materials-14-07119-f012:**
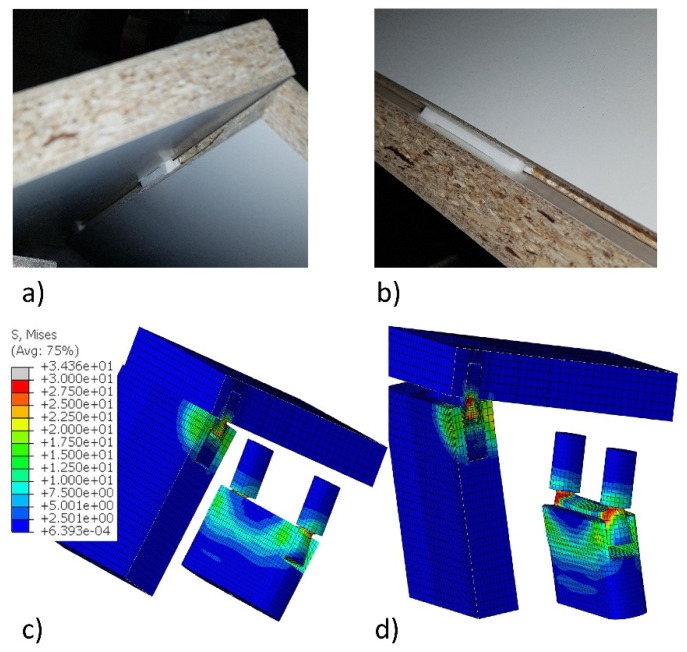
Actual failure modes and stress distribution of the joints: (**a**,**c**) under tension, (**b**,**d**) under compression.

**Table 1 materials-14-07119-t001:** Densities and some mechanical properties of the selected materials (* [36], ** [37]); k—bending strength, E—modulus of elasticity, G—shear modulus, ν—Poisson’s ratio, L,R,T—longintudinal, radial, and tangential direction.

Properties	Unit	Beech *	PL (UF) **	PB *	MDF *	HDF *	PA12
Density	kg/m^3^	734	798	642	745	891	938
$k_{L}^{b}$	MPa	95	89	12	32	57	26
$k_{R}^{b}$		43	38				
$E_{L}$		14,100	8636	2530	3850	5456	709
$E_{R}$		2280	2661				
$E_{T}$		1160					
$G_{LR}$		1645	822	987	1480		
$G_{LT}$		1082					
$G_{RT}$		471					
$\nu_{LR}$		0.450	0.439	0.282	0.300	0.300	0.229
$\nu_{LT}$		0.510					
$\nu_{RT}$		0.750					
$\nu_{TR}$		0.360					
$\nu_{RL}$		0.075	0.031				
$\nu_{TL}$		0.044					

**Table 2 materials-14-07119-t002:** Differences between the maximum loads for the results of experiments and numerical calculations. COV: Coefficients of variation.

Loading Type	Material Type	Maximum Load (N)			Differences (%)
		Experiment		FEM	
		Mean	COV (%)		
Tension	Be	212.89	8.75	212.89	0.00
	PL	193.14	3.39	198.55	−2.80
	HDF	114.56	8.81	115.96	−1.22
	MDF	86.76	10.86	89.56	−3.23
	PB	70.08	11.45	74.96	−6.96
Compression	Be	147.76	10.07	144.88	1.95
	PL	140.01	9.58	139.37	0.46
	HDF	77.71	10.98	75.10	3.36
	MDF	56.91	11.26	54.57	4.12
	PB	37.55	9.14	37.03	1.38

**Table 3 materials-14-07119-t003:** Summary of ANOVA results for bending moment capacity and stiffness.

	Source	Degrees of Freedom	Sum of Squares	Mean Squares	F–Value	*p*–Value
Tension						
Bending moment	Material type	4	551.29	137.822	318.15	0.000
	Error	45	19.49	0.433		
	Total	49	570.78			
Stiffness	Material type	4	147.770	36.9425	237.88	0.000
	Error	45	6.988	0.1553		
	Total	49	154.759			
Compression						
Bending moment	Material type	4	330.19	82.5469	233.00	0.000
	Error	45	15.94	0.3543		
	Total	49	346.13			
Stiffness	Material type	4	10.046	2.51157	106.60	0.000
	Error	45	1.060	0.02356		
	Total	49	11.106			

## Data Availability

The data supporting the reported results can be sent by the authors via e-mail.

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
