# Peer review of "Analysis of the Internal Mounting Forces and Strength of Newly Designed Fastener to Joints Wood and Wood-Based Panels"

_materials, 2021, doi:10.3390/ma14237119_

Round 1

Reviewer 1 Report

Well written article. Comments for improvement are as follows:

  1. The first sentence in the Abstract is not necessary.
  2. Keywords should be corrected. The keywords "experiment" and "FEM" are clichés and are not adequate.
  3. At the end of the Introduction section, emphasize the scientific innovation of your research.
  4. List briefly what has been done in each section.
  5. The universality of the methodology should be emphasized. Perhaps a flowchart could be displayed at the beginning of the "Materials and Methods" section. The flow chart of the methodology should be adequately described.
  6. The results discussion must be enlarged and more comparisons with literature data should be reported. The authors should go more in depth with the interpretation of the results.
  7. In the conclusions, add the limitations of the applied methodology and the directions of future research.

Author Response

Title: Analysis of the Internal Mounting Forces and Strength of Newly Designed Fastener to Joints Wood and Wood-based Panels

Łukasz Krzyżaniak*, Tolga Kuşkun, Ali Kasal and Jerzy Smardzewski*

Responses to the Editor and the Reviewer

Dear Editor and the Reviewers,

First of all, we want to thank you for offering the opportunity to respond to the reviewers' comments. Below, step by step, we present our explanations and changes introduced to the text. All changes (in manuscript) are marked in red.

Responses to Reviewer #1:

We are grateful to the Reviewer for careful reading of the manuscript and helpful remarks. Below we respond to the remarks one by one.

Q1: The first sentence in the Abstract is not necessary.

A1: The authors deleted the first sentence in the Abstract.

Q2: Keywords should be corrected. The keywords "experiment" and "FEM" are clichés and are not adequate.

A2: According to the Reviewer's suggestion, keywords were corrected.

Q3: At the end of the Introduction section, emphasize the scientific innovation of your research.

A3: At the end of the Introduction, scientific innovation of the research was presented.

In the context of presented literature, limited studies have raised the problem of relationship between the strength of furniture joints and internal forces resulting from the assembly of elements. Furthermore, there have also been no studies explaining how the geometry of the fasteners and the type of materials used for their production will affect the mounting forces. Consequently, the relationship between the mounting forces in the newly designed fasteners and the stiffness of these joints has not been established.

Q4: List briefly what has been done in each section.

A4: A short description of tasks in each section was introduced.

The disposition of this study could be expressed in 4 section. In section  1, the shape of the newly designed fastener, mathematical model and distribution of the internal forces were presented. In section 2, preparing of the L-type corner joints connected with this fastener was described. Section 3 and 4 include the presentation of testing and numerical analyses of the joints, respectively.

Q5: The universality of the methodology should be emphasized. Perhaps a flowchart could be displayed at the beginning of the "Materials and Methods" section. The flow chart of the methodology should be adequately described.

A5: At the beginning of the "Materials and Methods" section, the flowchart of the used methodology was added.

Figure 1 shows a flowchart illustrating the universality of the methodology used. In the first step, the engineer designs and analyzes the construction of fasteners, their geometry, and the possibility of modifying the shape. Next, the method requires collecting the basic elasticity constants of the materials from which the joints are made. The next step involves analytical modeling of the internal mounting forces. This leads to an explanation of where the desired interactions can be expected to occur. Then, the method provides for the parallel conduct of experimental and numerical tests to determine the strength and stiffness characteristics of the joints. Finally, the engineer should compare the relationship between the force and displacement numerically and experimentally measured. The high correlation of this relationship proves the correct calibration of the numerical model. Next, the engineer should analyze the mounting forces in the joint and the interaction between the surfaces of the fasteners. The last stage of implementing the described method is assessing the influence of mounting forces and the type of materials used on the stiffness and strength of furniture joints.

Q6: The results discussion must be enlarged and more comparisons with literature data should be reported. The authors should go more in depth with the interpretation of the results.

A6: Comparisons with the literature and some interpretations were added to the ‘Results and Discussion’ section.

The bending moment capacity results obtained from this study have been compared with the results in some similar studies in literature [4][21][24][27][28][32]. In these stud-ies, bending moment capacities of different kinds of corner joints constructed of PB or MDF were investigated by many researchers.

In the study carried by Jivkov et al.[4], bending moment capacity was obtained as 7.73 Nm for PB-minifix joints while it was obtained as 9.91 and 14.24 Nm for the same combi-nation in the studies of Šimek et al. [21] and Yerlikaya [24], respectively. In these studies, bending moment capacity value of glued-dowel and unglued dowel joints was 20.94 and 14.1 Nm. Accordingly, it is understood that the use of glue in the joints has a significant effect on the strength. According to the results of the studies investigating the screwed corner joints [27][28], it can be clearly seen that screw connections provide much higher bending moment capacity values than the other fasteners. The bending moment capacity of glued-screwed and un-glued screwed joints was given as 87.98 and 70.92 Nm in the mentioned studies. Here, it should be noted that, from the point of view of the engineering design approach, the strength of the joints is only related to the loads they must carry in service. The bending moment capacity values obtained in the study [32] in which two dif-ferent newly externally invisible fasteners were designed and their performance measured were 19.75, 19.53, 7.13, 7.40, 8.66 and 9.62 Nm for the joints PB-fastener1 (S-PB), MDF-fastener1 (S-MDF), PB-fastener2 (M-PB), MDF-fastener2 (M-MDF), PB-minifix (E-PB), and MDF-minifix (E-MDF), respectively.

When the bending moment capacity values obtained from these mentioned studies are compared with the bending moment capacity values of this study, it can be seen that there is consistency between the results. However, it is expected that there will be some differences between the bending moment capacity values due to the differences in L-type specimen sizes and corresponding moment arms. The fact that the results obtained from this study are compatible with different joining techniques tested in other studies is an in-dication that the newly designed fastener can be an alternative for the corner joints of case furniture.

It is clear that from the all those studies, generally beech and plywood yielded higher bending moment capacity results than the other wood based panels, similarly MDF showed higher results than PB, and screwed joints gave higher bending moment capaci-ties than the other joints tested. When the studies are observed in literature, it can be seen that bending moment capacities under tension and compression are in a linear relation-ship with the density of the materials used.

Q7: In the conclusions, add the limitations of the applied methodology and the directions of future research.

A7: In the conclusions, limitations of the applied methodology and the directions of future research was added.

The presented method applies only to joints in which mounting forces are generated. Therefore, its use in the case of adhesive joints will not be effective.

A new direction of research using the developed method will be the implementation of metamaterials with a negative Poisson's ratio for modeling increasing mounting forces in furniture joints.

Once again, the authors want to thank you for offering the opportunity to respond to the reviewer's comments. We sincerely hope that the answers are satisfactory and fully meet the reviewer's expectations.

Jerzy Smardzewski

Reviewer 2 Report

The authors must address the following list of issues before it can be accepted.

  1. unnecessary use of references and self references. for example in page 7 line 237, authors cite references from 38 to 43 for such an obvious statemente "The practice of numerical modeling using the finite element method shows that models should be as simple as possible and extremely match to the modeled real objects". this is improper and a bit unethical.
  2. authors must decide if they use "fig." or "figure"
  3. units are missing in figure 1.
  4. figure 1 needs labels as it shows too many things, which is the "tendon" and which the "mortise"? what are we looking at the bottom right?
  5. the 3d printing process needs more details, for example the printing orientation of the parts?
  6. figure 3 needs labels, what is shown on the left?
  7. Section 2.3, why is it called "tension"? it looks like a 3 point bending test. 
  8. figure 4 needs units, and further description in the caption, what is e' or e'', or f?
  9. finally, some thoughts on why one should 3D print a fastener?? should included in the conclusions

Author Response

Title: Analysis of the Internal Mounting Forces and Strength of Newly Designed Fastener to Joints Wood and Wood-based Panels

Łukasz Krzyżaniak*, Tolga Kuşkun, Ali Kasal and Jerzy Smardzewski*

Responses to the Editor and the Reviewer

Dear Editor and the Reviewer,

First of all, we want to thank you for offering the opportunity to respond to the reviewers' comments. Below, step by step, we present our explanations and changes introduced to the text. All changes (in manuscript) are marked in red.

Responses to Reviewer #2:

We are grateful to the Reviewer for careful reading of the manuscript and helpful remarks. Below we respond to the remarks one by one.

Q1: Unnecessary use of references and self references. for example in page 7 line 237, authors cite references from 38 to 43 for such an obvious statemente "The practice of numerical modeling using the finite element method shows that models should be as simple as possible and extremely match to the modeled real objects". this is improper and a bit unethical.

A1: This important aspect of numerical modeling has been repeatedly discussed in numerous cited publications. Unfortunately, there are many examples of overly exaggerating the entanglement of the finite element method for simple experiments. Nevertheless, the results of such experiments do not differ from sufficiently simplified models. Therefore, at the request of the Reviewer, the paragraph in question was omitted.

Q2: authors must decide if they use "fig." or "figure"

A2: The authors replaced the "Fig." with the term "Figure" in the whole text.

Q3: units are missing in figure 1.

A1: All dimensions in mm and the unit was added to the related Figure caption.

Q4: figure 1 needs labels as it shows too many things, which is the "tendon" and which the "mortise"? what are we looking at the bottom right?

A4: The description of Figure 1 was improved.

Q5: the 3d printing process needs more details, for example the printing orientation of the parts?

A5: An explanation related to the 3D printing process was added to the ‘Preparation of the L-type corner joints’ section

During the 3D printing, the thickness of layers was equal 0.2 mm. Each layer was applied with an infill density of 100% towards to the long side of fasteners.

Q6: figure 3 needs labels, what is shown on the left?

A6: The caption of this Figure was improved.

L-type corner joint specimens constructed of: a) MDF, b) HDF, c) PL, d) PB, e) Be.

Q7: Section 2.3, why is it called "tension"? it looks like a 3 point bending test.

A7: The terms "tension" and "compression" under joint tests are commonly used in the literature. However, using a different term would be confusing to readers, and the reviewers would undoubtedly suggest a change of terminology.

Q8: figure 4 needs units, and further description in the caption, what is e' or e'', or f?

A8: The description of Figure 4 has been improved.

Q9: finally, some thoughts on why one should 3D print a fastener?? should included in the conclusions

A9: The Reviewer's suggestion was included in the conclusions.

The research has shown the usefulness of 3D printing for the production of experimental fasteners for furniture joints. In industrial practice, however, it is proposed to use cheaper injection technology.

Once again, the authors want to thank you for offering the opportunity to respond to the reviewer's comments. We sincerely hope that the answers are satisfactory and fully meet the reviewer's expectations.

Jerzy Smardzewski

Round 2

Reviewer 1 Report

The article has been corrected and supplemented. I recommend accepting the article in its current form.